# Transdermal Drug Delivery in the Pig Skin

**DOI:** 10.3390/pharmaceutics13122016

**Published:** 2021-11-26

**Authors:** Ignacio Ordiz, José A. Vega, Raquel Martín-Sanz, Olivia García-Suárez, Miguel E. del Valle, Jorge Feito

**Affiliations:** 1Departamento de Morfología y Biología Celular, Universidad de Oviedo, 33006 Oviedo, Spain; correo@ordizmesoterapia.com (I.O.); javega@uniovi.es (J.A.V.); garciaolivia@uniovi.es (O.G.-S.); miva@uniovi.es (M.E.d.V.); 2Grupo SINPOS, Universidad de Oviedo, 33006 Oviedo, Spain; 3Facultad de Ciencias de la Salud, Universidad Autónoma de Chile, Providencia, 7500912 Santiago de Chile, Chile; 4Servicio de Oftalmología, Complejo Asistencial Universitario de Salamanca, 37007 Salamanca, Spain; rmartinsan@saludcastillayleon.es; 5Servicio de Anatomía Patológica, Complejo Hospitalario Universitario de Salamanca, 37007 Salamanca, Spain

**Keywords:** transdermal delivery, intradermal injection, mesotherapy, microneedling, electroporation, skin, morphology

## Abstract

Transdermal delivery can be accomplished through various mechanisms including formulation optimization, epidermal stratum corneum barrier disruption, or directly by removing the stratum corneum layer. Microneedling, electroporation, a combination of both and also the intradermal injection known as mesotherapy have proved efficacy in epidermal-barrier disruption. Here we analyzed the effects of these methods of epidermal-barrier disruption in the structure of the skin and the absorption of four compounds with different characteristics and properties (ketoprofen, biotin, caffein, and procaine). Swine skin (Pietrain x Durox) was used as a human analogue, both having similar structure and pharmacological release. They were biopsied at different intervals, up to 2 weeks after application. High-pressure liquid chromatography and brightfield microscopy were performed, conducting a biometric analysis and measuring histological structure and vascular status. The performed experiments led to different results in the function of the studied molecules: ketoprofen and biotin had the best concentrations with intradermal injections, while delivery methods for obtaining procaine and caffein maximum concentrations changed on the basis of the lapsed time. The studied techniques did not produce significant histological alterations after their application, except for an observed increase in Langerhans cells and melanocytes after applying electroporation, and an epidermal thinning after using microneedles, with variable results regarding dermal thickness. Although all the studied barrier disruptors can accomplish transdermal delivery, the best disruptor is dependent on the particular molecule.

## 1. Introduction

The ease of use of topical presentations has made the skin target of different cosmetic and/or “therapeutic” methods from time immemorial [1]. Skin represents the biggest organ of the human body, and has several functions regarding touch, defense, and hydration maintenance. All these functions require a formidable superficial barrier to be sustained against chemical and physical agents, behaving as an impediment for topical delivery of most substances, although small-size lipophilic drugs may still reach therapeutic levels with topical administration, demonstrating that the barrier is actually selective [2]. It is in turn composed of various barriers working synergistically, including the stratum corneum, epidermal tight junctions, basement membrane, and blood vessel endothelia; these barriers can be found in the epidermis, but also in skin adnexa [3].

The most external barrier is called stratum corneum or horny layer, where cells have completed the keratinization process initiated in the basal epidermal layer, losing their nuclei and acquiring lipophilic properties [4] in a cycle of up to approximately 1 month [5]. The lipidic predominance is particularly found in the extracellular space of the stratum corneum, although all the epidermis behaves this way [6]. This lipophilic property is also associated with dehydration in the stratum corneum, where water constitutes approximately 15–20% of the total mass, whilst in other living epidermal layers it constitutes as much as 70% [7].

Epidermal tight junctions are the second cutaneous barrier, found mostly in the stratum granulosum, where they form a bidirectional barrier, active for most molecules including water [3,8]. These junctions are composed of various proteins, mostly of the claudin family, which is also responsible of most of the tight-junction barrier functions [3,8,9].

Between the epidermis and dermis is found a basal membrane composed of various proteins, mainly laminins and collagens, mostly of type IV, with collagen VII playing a critical role in the anchor to the dermal extracellular matrix and epidermal hemidesmosomes [10,11]. This structure plays a capital role as both a mechanical supporter and regulator of the epidermal metabolic influx and it is also transited by leukocytes and Langerhans cells [12].

The superficial vascular plexus is the deepest skin barrier [9], encompassing superficial blood and lymphatic vessels. It is found among the papillary dermis and the more collagenized upper reticular dermis. The connective dermal tissue surrounding the plexus may demonstrate striking degeneration due to ageing and solar exposure, changing its composition and physical properties [13].

Cutaneous permeation of different drugs, also known as transdermal (or more properly transepidermal) delivery, leads to an easy, quick, and efficient pharmacological action. Another advantage is that applied dosage does not decrease after washing, which is what happens in simple topical or patch administration [14]. To accomplish topical permeation of pharmacological agents, permeating agents or enhancers have been developed for use on one side (formulation optimization) and epidermal disruptors on the other side (actively manipulating the epidermal barrier function). These can even be combined [4,15,16]. The most important penetration factors are lipophilicity, molecular size, and employed excipients [17].

Epidermal disruptors are a group of active techniques aimed at overcoming the epidermal barrier. They include intradermal injections (ID), iontophoresis, electroporation (EP), and microporation employing different microneedle arrays (MN), which can even be jointly administered [4,18]. ID or mesotherapy directly introduces medication through a needle in the dermal tissue, as close as possible to the zone where the pathology is located [19]. Its performance varies with the particular drug employed and the exact depth of injection [20], and also increases with multiple injections, thus fragmenting the dose [21]. ID has demonstrated superior efficacy and immediate effects compared to hypodermal injection regarding insulin administration [22,23], vaccination [24,25], and topical anesthetics [26,27]. Furthermore, the ability of applying sterile water though ID for pain relief is widely known [28,29,30]. It has even demonstrated similar efficacy to laser therapy in the treatment of melasma [31].

EP generates small nanometric orifices in the stratum corneum under an electric field with variable temporal intervals, creating brief disruptions in the epidermal lipophilic barrier, and in this way dodging the barrier [4]. The electric field also helps to “push” different electrically charged molecules through the pores, and for this reason delivery efficacy is related to the specific electrical properties of the molecules being delivered and to the electrical field [18]. Generally speaking, the electrical field efficacy depends on the amount of energy transferred, bringing together pulse frequency, pulse duration, and voltage [32,33]. The pulses can even change over time, modifying the delivery efficacy [34]. Probably the main drawbacks of employing electric fields are the temperature increase [35], muscle contractions, and pain [36]. Cellular damage caused by polar effects of the applied electrodes has also been reported [37].

EP drawbacks can be avoided by combining it with iontophoresis, promoting drug penetration while maintaining a low level of adverse effects [38,39]. Iontophoresis enhances cutaneous delivery of charged substances by means of small intensity currents, creating a collateral water flow (electroosmosis) that can also drive neutrally charged molecules through the horny layer [40]. The efficacy of this combination in skin permeation has already been demonstrated [41] and there are excellent perspectives for this technique in oncologic therapies and vaccine delivery [42].

MN employs microneedles, generating epidermal micropores as an initial permeator and applying a topical pharmaceutical solution which penetrates through the generated pores; variants include the use of hollow or dissolvable needles or application of the medicament over the needles (coating) before performing the microneedling [16]. The micropores generated this way avoid the horny layer and have the advantage of a standardized diameter and depth of penetration. The greater the diameter (via thicker needles) and the shorter the pores (via previous abrasion of the epidermis), the higher will be the available dose [43], an observation confirmed in commercially available microneedles [44]. Fabrication of the microneedles is complex, with varied forms and materials [45,46,47]. Just as with other transdermal routes, MN is also effective in vaccine delivery in patches [48] or in a single application [49], even employing dissolvable microneedles [50]. MN patches are currently used in the delivery of various active substances including nicotine, some hormones, some anesthetics, anti-inflammatory medications, nitroglycerin, and scopolamine, giving precise dosing and few systemic complications [2,51,52]. MP, particularly employing dissolvable microneedles [53], can be associated with nanocarriers [44,54], improving the effect of cancer chemotherapy [55].

The technique can be combined with other epidermal disruptors to enhance their effect [18,56]; for example, a combination between EP an MN have previously been demonstrated to facilitate transepidermal delivery of hydrophilic macromolecules without producing cutaneous damage in swine skin [57].

MN, ID, EP, and MN + EP were chosen to be experimentally compared, performing skin biopsies to measure the microscopical findings and perform high-pressure liquid chromatography (HPLC) as the method to measure the efficacy of transdermal delivery [58].

## 2. Materials and Methods

### 2.1. Substances

To conduct the experiment, four different active principles were chosen to reflect possible variations of the transdermal permeation mechanisms regarding the different molecular properties.

There was employed 4 mL of ketoprofen in the injectable presentation Orudis^®^ (Sanofi-Aventis, Paris, France). The active ingredient was ketoprofen at a concentration of 50 mg/mL, with benzyl alcohol as excipient.

Biotin was employed in the injectable presentation, Medebiotin^®^ (Medea, Barcelona, Spain) and 4 mL was injected from 5 mL vials. The concentration of sodium biotin was 5 mg/mL, equivalent to 4.6 mg of pure biotin. Other components were sodium hydroxide and water.

Procaine was also employed in an injectable presentation (Serra Pàmies, Reus, Spain) and 4 mL was injected from 5 mL vials at a concentration of 10 mg/mL. Excipients were chlorohydric acid for pH adjustment and water.

Caffeine was produced by the laboratory Mesoestetic (Barcelona, Spain) in a formulation specific for ID. The composition was 3% caffeine, 3% sodium benzoate, and 94% milliQ csp 100 water.

### 2.2. Animal Model and Experimental Procedure

Six crossbred (Pietrain × Duroc) 10-weeks-old pigs were acquired for experimentation. They were four females and two males, with weights between 35 kg and 38 kg. Before performing the described experimental procedures, the pigs were sedated and anesthetized with intramuscular azaperone (2 mg/kg), ketamine (10 mg/kg), and xylazine (0.08 mg/kg). Later, a subcutaneous injection of 0.01 mg/kg buprenorphine was applied. Then, the loins of the pigs were shaved and washed and antisepsis was performed with iodized povidone. Finally, a peripheral line was placed in the auricular vein to infuse physiological saline solution, propofol, and isofluorane.

Both loins of each pig were divided in 10cm × 10 cm quadrants and each of the four techniques in the study were applied on one them (Figure 1a,b). Then, each quadrant was subsequently divided in another four small 5cm × 5 cm quadrants which corresponded with different time lapses after drug appliance, identified as T0, T15, T30, and T60 expressing the minutes after application (Figure 1c,d).

The different substances were applied with different techniques over the quadrants, defining time zero (T0) as the time of completion of each technique. Then, they were sampled with an 8 mm punch to sample epidermis, dermis, and superficial hypodermis, at intervals of 15 (T15), 30 (T30), and 60 (T60) minutes after time zero. Two biopsies were taken in each quadrant (Figure 1e), one fixed in formaldehyde and a second one to be frozen at −40 °C. Wounds were sutured and washed preserving antisepsis (Figure 1f); three layers of Nobecutan^®^ (Inibsa, Barcelona, Spain) were applied to form a protective plastic film. In addition, prophylactic penicillin–streptomycin was administered for three consecutive days.

Pigs were returned to the farm after healing the wounds, and monitored to rule out possible toxicity or cutaneous hypersensitivity, following the recommendations of the American Association for Laboratory Animal Science [59,60]. Animals were returned again to Oviedo University to be sampled at days 7 and 14 after the initial experiment, returning to the farm thereafter. One animal could not be returned, as it died before the seventh day. Experimentation procedures were approved by the regional government of Principality of Asturias (code PROAE 09/2014) and manipulation was performed according to the recommendations of Birmingham University [61].

### 2.3. Microporation

The employed microneedle had a 6-needle basculation head, with a needle length of 1.5 mm, a 0.2 mm diameter, and a 1.5 mm separation between needles (MT.DERM Gmbh, Berlin, Germany). The oscillation (shaker) speed was set at 150 rpm, with a depth of 0.8 mm. MN time was limited to 10 min. During the MN process the 10 × 10 cm quadrant was irrigated with 5 mL of each substance. The procedure efficacy was proven by the appearance of erythema and small petechiae.

### 2.4. Electroporation-Iontophoresis

The employed equipment (Mesoestetic Medical Devices, Viladecans, Barcelona, Spain) was programmed to use with the Spanish electricity supply, employing 50 Hz frequency in alternate current. Application time was limited to 15 min. The formulation was introduced in a plastic recipient coupled to a steel roll-on mechanism, connected to the electroporation unit. The unit manufacturer also prepared the substances in gelified presentation it was applied in circular movements though the roll-on mechanism over the corresponding 10 × 10 cm quadrant. Both substance polarity and pH were considered, to improve cutaneous penetration as much as possible (Table 1).

### 2.5. Intradermal Injection

The technique was performed according to the indications of Chos [62], considering that the optimal depth of injection is half of the skin fold. The fold was measured with the help of a digital caliper (Würth, Künzelsau, Germany) to ensure intradermal delivery [61]. The volume of administered substance in each point was calculated in ml, dividing the depth of injection expressed in mm by 20, according with the American Association for Laboratory Animal Science [60].

A dose of 4 mL was uniformly divided in the 10cm × 10 cm quadrant. The intradermal injection was undertaken with a DHN3^®^ pistol (Techdent, France) including a stabilized baseplate. This device can regulate both the depth of injection and the dose fragmentation.

### 2.6. High-Pressure Liquid Chromatography (HPLC)

Frozen cutaneous samples were homogenized in 4 cc of 0.9% physiological saline solution in an Ultra-Turrax T8 equipment (Ika Labortechnik, Staufen, Germany) for 1 min at 20,000 rpm. The material was analyzed with an UPLC Dionex Ultimate 3000^®^ device (Thermo Scientific, Waltham, MA, USA) coupled to a mass spectrophotometer Impact II (Bruker, Billerica, MA, USA).

Detection was performed in a positive range of 100–800 Da for biotin, procaine, and caffeine, and in negative mode for ketoprofen. The electrospray source had a voltage of 4500 V, nebulization flux of 35 psi, and temperature of 250 °C. The specific parameters of the different substance dilutions are summarized in Table 2. All the samples were centrifuged at 10,000 rpm for 5 min.

### 2.7. Histological Assesment

The 12 most-representative samples were chosen based on HPLC results, including all the techniques and substances. The times T60, D7, and D14 were chosen as most representative since the histological events needed some time to develop even the most immediate changes (T60).

Skin samples were routinely processed in paraffin blocks and sectioned in 10-µm-thick sections. They were stained with hematoxylin–eosin and Masson trichrome.

### 2.8. Indirect Immunohistochemistry

The technique was performed employing the paraffin tissue, also cut in 10-µm-thick sections. Primary antibodies were directed against S100 protein (Dako, Glostrup, Denmark), Vimentin (Boehringer-Mannheim, Mannheim, Germany), and Collagen type IV (Santa Cruz Biotechnology, Dallas, TX, USA). To reveal the primary antibodies, we employed the simple peroxidase–antiperoxidase EnVision^®^ kit (Dako, Glostrup, Denmark), which includes the secondary antibodies. Immunohistochemistry was manually revealed using diaminobenzidine under a 40x objective in an Olympus BX23 microscope.

### 2.9. Quantitative Analysis of the Skin

Cutaneous thickness was measured employing the T0, T60, D7, and D14 samples. The measurement was performed over the Masson trichrome stain (with high contrast as the collagenized tissue is stained in blue). The material was observed and photographed in an Olympus BX61^®^ microscope coupled to an Olympus DP70^®^ 12 Mpix camera (Olympus, Tokyo, Japan), employing a 4× and also a 20× UPlanSApo objective.

To determine the thickness, algorithms and criteria proposed in the literature were followed [63,64]. Briefly the area of epidermis and dermis was manually defined in order to measure their thickness (Figure 2a–c). A mosaic of images was performed to build a super-image with the software Visiopharm^®^ CAST v2 (Hoersholm, Denmark). The software is able to reduce the marked area to different-diameter spheres that can be easily measured (Figure 2d–i).

The density of blood vessels, fibroblasts, melanocytes, and dendritic cells was investigated in five different sections per experiment and over a defined time; they were cut from the same paraffin block and were separated by at least 100 µm. Each section was scanned with a SCN400F scanner (Leica Biosystems™, Newcastle, UK) and annotated using Slide Path Gateway LAN software (Leica Biosystems™). A 4 mm^2^ digital grid was randomly applied onto the scanned images, and the structures and cells within the grid were separately counted by two observers (J.A.V. and I.O.) in a total of five non-overlapping fields. Both the observers and the samples were blinded. Results are expressed as the mean value of each measurement per mm of skin.

For the measurement of blood vessel and dermal fibroblasts, the number of circles identifiable as capillaries up to 1 mm below the dermo–epidermal junction was independently counted by two of the authors (J.A.V. and I.O.) in sections stained with hematoxylin–eosin and Masson trichrome and immunostained for Vimentin, totaling an area of 2 mm^2^ in the described grid.

The measurement of melanocyte and dendritic cell density was performed in sections immunostained for detection of S100 protein and Vimentin in the scanned images. Epidermal S100 protein-positive cells were first counted, and the Vimentin-positive cells were counted thereafter. To calculate melanocytes, the number of S100+ cells was employed and to calculate dendritic cells the number of S100+ cells was deducted from the total Vimentin+ cells.

## 3. Results

### 3.1. Histological Findings

Cutaneous samples taken from the back of different pigs were histologically very similar to other mammalian species non-glabrous skin regarding their structure (Figure 3a). There are subtle anatomical differences with human skin regarding skin adnexa—although eccrine glands predominate in pig dorsal skin, apocrine glands, which are not found in human skin of the back, could be seen on occasion (Figure 3b).

Dermal capillaries are probably the final structure regarding transdermal delivery, and a superficial dermal plexus was readily found in the boundary between papillary and reticular dermis (Figure 3c,d).

One of the most remarkable data of the study is that none of the molecules demonstrated evident histological alterations under the widefield microscope. The observed anomalies could be related with the puncture, generally affecting the epidermal epithelium and in to lesser degree, the superficial dermis (Figure 4). We observed no significant leukocyte component morphologically in the studied samples, dismissing inflammatory changes.

### 3.2. Epidermal Dendritic Cells

Regarding the dendritic cell populations of the epidermis, we were able to discern melanocytes from Langerhans cells in the pig skin through immunohistochemistry for S100 and Vimentin. In our hands, the manually-performed S100 protein technique was capable of demonstrating melanocytes, and Vimentin indicated Langerhans cells. The S100 protein and Vimentin-positive cells located in the epidermal basal portion were counted as melanocytes, and those positive for Vimentin in an upper position within the squamous or granular layers were assumed as Langerhans cells (Figure 5). The immunohistochemical profile of swine skin, the with absence of S100 protein reaction in the Langerhans cells, is concordant with previous results [65].

Both cell types were quantified in the different samples according to the different disruption procedures, revealing higher counts of either melanocytes or Langerhans cells in EP-treated pigs, including the MN + EP-treated specimen (Table 3).

The results demonstrate an increase in both densities in nearly all the studied samples. The highest increase both melanocyte density and Langerhans cell density was found in the pigs treated with EP, with MN + EP in second place (Table 4).

### 3.3. Morphometrical Analysis

To perform skin-thickness measurements, samples obtained after one (D7) and two (D14) weeks were chosen, with the initial biopsy (T0) as control. No relevant changes were devised in the first hour after treatment. The analysis was performed in the MN and MN + EP samples, because no histological changes were expected after a single injection nor in EP where there was no dermal aggression.

A reduction in epidermal thickness was observed with the MN technique in the three differently treated animals, while an epidermal-thickness increase was observed in two of the MN + EP treated animals (Table 5). Regarding dermal thickness, there was an evident increase in two of the MN-treated animals, while these animals demonstrated a subtle increase when employing the MN + EP technique; the remaining animal showed little reduction in both measurements (Table 5).

To assess the blood vessel area and fibroblast density after applying the different techniques, Vimentin immunostains were used and a manually-defined area was chosen for comparison (Table 6).

The results demonstrate minimal differences in vascular density both at the first hour (T60) and the final stages (D14) of the experiment. However, there was an increase in the vascular area after a week (D7) of administration, observed particularly when employing the MN + EP technique, and with ID and MN to a lesser extent (Table 7). Differences regarding fibroblast density were subtle (Table 7).

### 3.4. High-Pressure Liquid Chromatography (HPLC)

All the techniques demonstrated higher dermal concentrations in the first 60 min of the experiment, whilst after one week, substances remained detectable at residual levels.

The most effective delivery technique in the animal treated with procaine was the MN + EP (Figure 6). This technique showed the highest peak and also the most sustained concentration over time. In this procaine comparison, the MN + EP combination was able to improve EP and MN pharmacological effects, although overall differences were limited (Figure 7 and Figure 8).

In the case of ketoprofen there was an obvious advantage for ID after the initial administration, with the highest concentration obtained after 15 min (Figure 6 and Figure 7). An unexpected finding was that the combination of MN and EP gave poorer results that either technique alone (Figure 8).

Biotin provided comparable results to ketoprofen, with an advantage for ID, showing the highest concentration after 15 min (Figure 6 and Figure 7). In this way, ketoprofen doubled the area under the curve of the next-most-effective technique, the MN (Figure 8).

Finally, the animal treated with caffeine offered contradictory results, with the highest concentration obtained when employing ID immediately after the administration. Concentration employing ID quickly diminished, and the best sustained concentration after the first hour of administration was achieved with the MN + EP technique (Figure 6). However, differences found in caffeine concentrations were limited (Figure 7 and Figure 8). Calibrations and result curves of HPLC are included In the Appendix A.

The performed experiments led to different results regarding the studied molecules– ketoprofen and biotin achieved the best concentrations with intradermal injections, while delivery methods for obtaining procaine and caffeine maximum concentrations changed in basis of the lapsed time, demonstrating little advantage for the MN + EP technique in the case of procaine, and for ID in the case of caffeine (Figure 7). However, the area under the curve showed a small advantage for the MN technique in the case of caffeine (Figure 8).

## 4. Discussion

Swine skin is one of the best animal models for studying human skin, particularly on pharmacological grounds, having similar structure and pharmacological release [65,66,67,68] and matched turnover periods [4], immunological properties [69], and wound-healing mechanisms [70]. However, differences exist between both species regarding pigmentation, age-related changes, elastic fiber composition, and skin adnexa distribution (mainly apocrine in pig while mainly eccrine in human) [65,66,67]. Regarding human skin, it is well known that cutaneous delivery may be influenced by several factors including age, sex, anatomical region, hydration degree, possible peeling of the skin, or even by ethnicity [71,72,73,74,75]. In the case of swine skin, there are also noticeable differences between different pig breeds, the domestic pig having been described as a comparable model [65]. This swine model has been previously considered suitable for the study of ketoprofen pharmacokinetics [76].

Our experiments demonstrated no obvious morphological changes attributable to epidermal disruption, except those related to pore generation (i.e., epidermal basal membrane discontinuity and focal bleeding). This way we can rule out drug-delivery-related inflammation or scar formation with the employed techniques. Our findings differ from those of other authors who describe marked skin thickness increase (between 140% and 650% in the epidermis) [77,78]. These differences were found in rats, and may be attributed to the different drugs and protocols employed. Nevertheless, the presented findings are in line with other authors, who have found no significant histological alterations after these procedures, including EP [32]. Other authors have only identified increased permeability and blood vessel constriction [79], probably due to EP-related disruption of endothelial cellular junctions [80]. Our results showed variation regarding a potential EP-associated epidermal thickening and also regarding a potential MN-associated dermal thickening. The MN-associated dermal thickening is supported by the mentioned increase in the vascular area of this specimen. Both findings have been previously described as specifically associated with the MN procedure [81], inducing tissue regeneration [82,83].

We were able to detect an increase in the number of dendritic cells in the EP-treated specimens, identified as melanocytes or Langerhans cells, which could be consistent with the finding of epidermal thickening. Melanocytes are responsible of cutaneous pigmentation and Langerhans cells are antigen-presenting cells, essential in the immune response [84]. Both cell types demonstrated proliferation associated with the EP procedure in our experiment. The melanocyte proliferation is scarcely found in the literature, although hyperpigmentation has been described in intradermal and intramuscular EP [85]. The presence of an increased number of Langerhans cells may be related with the reported success of EP in the delivery of vaccines [15,42]. Langerhans cells respond to physical stimuli proliferating and migrating from the skin to lymphatic ganglia, as if there had been an antigen presentation [86]. This mechanism can explain the faster and more effective immune response of transdermal EP delivery compared to the intramuscular route [87].

Even though skin constitutes a formidable barrier for all chemical elements (particularly hydrophilic ones), its advantages over other routes are minor toxicity, absence of gastric or renal complications, and the avoidance of hepatic metabolism, preventing possible interactions with other drugs [88]. In this respect, transdermal delivery of ketoprofen resulted in a 10-fold reduction of gastric ulcers compared with oral delivery [89], while up to 20-fold higher macromolecule absorption than hypodermal delivery has also been proven [90]. Dermal microcirculatory and lymphatic features were invoked to explain this phenomenon [91].

The studied techniques have demonstrated, thanks to the HPLC experiment, their ability to permeate the hydrophobic stratum corneum barrier and to cross the hydrophilic living epidermis to reach the dermal blood vessels. Substances can reach the systemic circulation though the blood, but they can also anchor to the extracellular matrix, cellular membranes, or even interact with peripheral nerves. These interactions with dermal constituents may extend the drug half-life in the dermis, justifying the detection of all the studied molecules after two weeks at residual levels, a fact already mentioned from a histological standpoint [92], but with unknown biological effect. The sustained presence of molecules has the advantage of a putatively increased therapeutic action over time.

Although the studied epidermal disruptors can accomplish transdermal delivery, the best disruptor is dependent on each particular molecule applied. Each permeabilization method is best suited for certain applications [18], and for this reason a precise knowledge not only of the physical and chemical features of the different compounds, but also of their biological properties is mandatory [93]. In our experiment, the marked differences in the cutaneous concentrations of the different molecules may be explained on the basis of their different dilutions and chemical properties.

In our case, differences were noticeable regarding ketoprofen and biotin administration, favoring ID in both cases. MN + EP or EP techniques were slightly favored regarding procaine in our experiment, followed by ID. The ability of ID to deliver ketoprofen and procaine was described 40 years ago by Binaglia [94,95]. Intradermal ketoprofen has already demonstrated efficacy in the treatment of acute pain in carpal tunnel syndrome [96]. This substance can be also delivered through microneedle arrays [97] and iontophoresis [98]. Regarding caffeine, there was a visible advantage for ID in the first minutes after administration, but differences were small after the first 15 min.

Differences in cutaneous drug concentration can be interpreted according to the depth of application, and are highlighted when comparing ID with MN, favoring the former in almost all the performed measurements of our experiment. It has already been described that minimal variations in the depth of injection lead to great changes in the pharmacokinetics of macromolecules [90], probably because ID achieves closer delivery to the vascular plexus and higher absorption rate. The dosage fragmentation of intradermal delivery also results in an increased uptake, increasing the potential of this technique [21]. Furthermore, significant differences when employing different drug formulations with the same procedure have been noted [44], so more experiments are required to clarify the real efficacy of these delivery methods.

Integrating all the results, ID achieved the best pharmacological concentrations in our experiment, but its main drawback was its dependency on the injector’s ability. The literature provides detail of many mesotherapy complications, generally related to granulomas and cutaneous mycobacteriosis [99,100]. For this reason, MN is considered a simple and less painful way to increase topical absorption, representing a delivery method with great potential for many other molecules [53,101]. Micropores generated though MN have been estimated to be maintained for 15 min [81,102], although their kinetics are delayed in older subjects [103], and occluding the microperforated area can extend the operative period of the pores up to 48–72 h [104]. This technique is suitable to quickly (5–20 min) enable a relatively high-quantity (up to 1.5 mL) transepidermal delivery [105]. Our study indicates that EP can improve the efficacy of MN when applying certain substances, while in other cases EP alone achieved better results.

## 5. Conclusions

The skin can be an effective route for drug delivery, particularly if its permeability is increased though ID, EP, MN, or EP + MN. Transdermal delivery has a long road to progress, and pharmacological innovations will bring this route to a broader range of indications [15]. These techniques produce minor histological alterations after their application, except for an epidermal thinning and dermal thickening associated with an increased vascular density after applying MN, and an increase in Langerhans cells and melanocytes after applying EP.

## Figures and Tables

**Figure 1 pharmaceutics-13-02016-f001:**
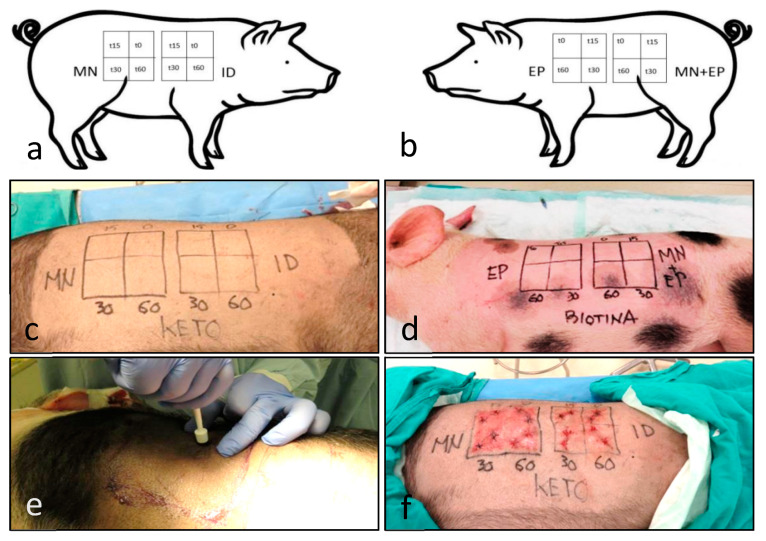
Design of the experimental procedure in the pigs. The working diagram of (**a**) the right side and (**b**) the left side) shows the different techniques and time intervals. The diagrams were applied to each of the differently treated animals. The images show the ketoprofen-treated pig, labeled as KETO (**c**) and the biotin-treated animal, labeled as BIOTINA (**d**). Punch biopsies were performed according to the diagram (**e**), and then sutured (**f**). Legend for quadrants: microneedle (MN), intradermal injection (ID), electroporation (EP) and the combination MN + EP; subquadrants depict the time after drug administration in minutes (0, 15, 30, and 60 min).

**Figure 2 pharmaceutics-13-02016-f002:**
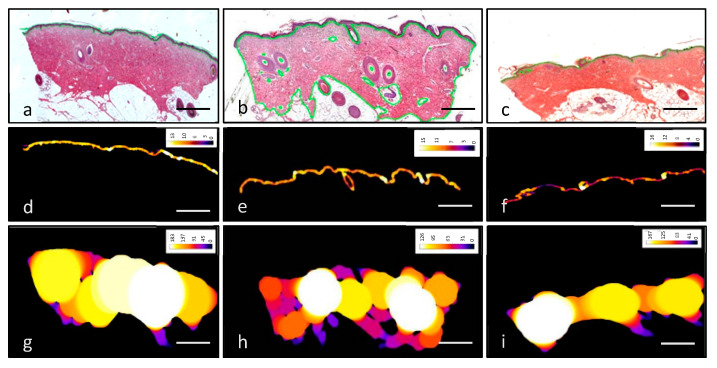
Method of measurement of epidermal and dermal thickness. Skin samples were stained with hematoxylin–eosin and prepared to be measured through the analysis software (**a**–**c**). Images were processed to quantify epidermal (**d**–**f**) and dermal (**g**–**i**) thickness by reducing the area to spheres. Colors in images d–i refer to the diameter of the spheres, which is depicted in the color legend in the upper-right-hand corner of each picture, measured in µm. Scale bar 1 mm.

**Figure 3 pharmaceutics-13-02016-f003:**
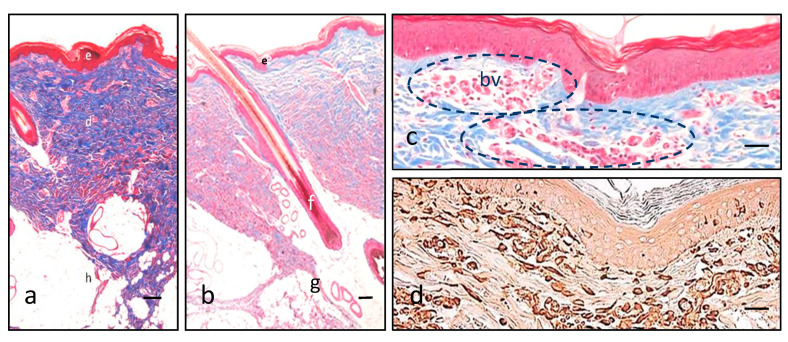
Features of pig skin. Masson trichrome (**a**–**c**) illustrates the three main layers in different colors with low magnification (epidermis in red, e; dermis in blue, d; hypodermis in white, h), specifically coloring collagen fibers in blue (**a**). Hair follicles (f) are not distinguishable from human, but closely there are some apocrine glands (g) that help in this distinction (**b**). A superficial vascular plexus is readily identifiable in red with Masson trichrome (blood vessels, bv, and dashed circles), highlighted amongst the bluish collagen background of the dermis (**c**). Vimentin intermediate filaments are demonstrated in brown due to the diaminobenzidine, and are typically present in epidermal dendritic cells and dermal fibroblasts and endothelial cells of the blood vessels (**d**). Scale bar 200 µm (**a**,**b**), 80 µm (**c**,**d**).

**Figure 4 pharmaceutics-13-02016-f004:**
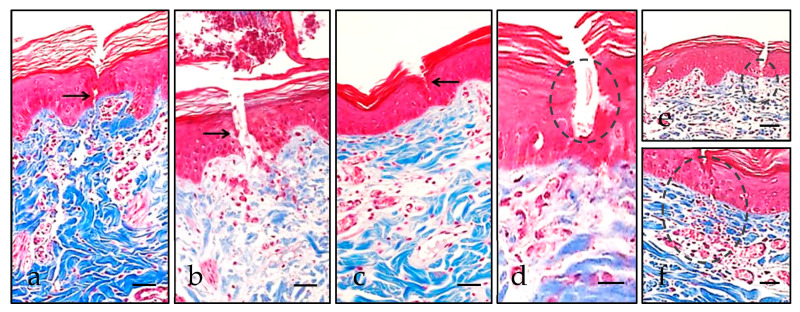
Microscopic lesions caused by puncture. Epidermal solutions of continuity, highlighted with black arrows, related to ID (**a**) were morphologically deeper than the MN-related ones (**b**,**c**). Some erythrocytes (highlighted with dashed circles) can be identified over the cutaneous disruption (**d**), at the dermo–epidermal junction (**e**), or even in the superficial dermis (**f**). Scale bar 40 µm (**a**–**c**,**e**,**f**), 20 µm (**d**).

**Figure 5 pharmaceutics-13-02016-f005:**
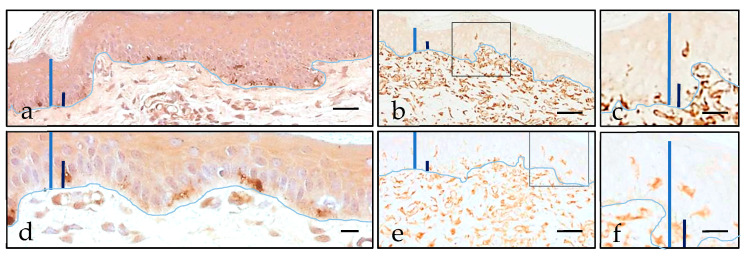
Immunohistochemical location of epidermal dendritic cells after 7 days (**a**–**d**) and after 14 days (**e**,**f**). Epidermis thickness is highlighted with a blue bar, epidermal basal cell layer thickness is highlighted with a dark blue bar, and the dermo–epidermal junction is outlined by a clear blue line. The highest density of melanocytes was observed in the EP specimen, where S100 protein demonstrated a slightly dendritic population in the epidermal basal layers (**a**,**d**). Langerhans cells were also readily apparent above the basal cell layer in the Vimentin-immunostained EP specimen (**b**,**c**,**e**,**f**). Images c and f are details of the squared regions in images b and e. Scale bar 40 µm (**a**,**b**,**d**,**e**), 20 µm (**c**,**f**).

**Figure 6 pharmaceutics-13-02016-f006:**
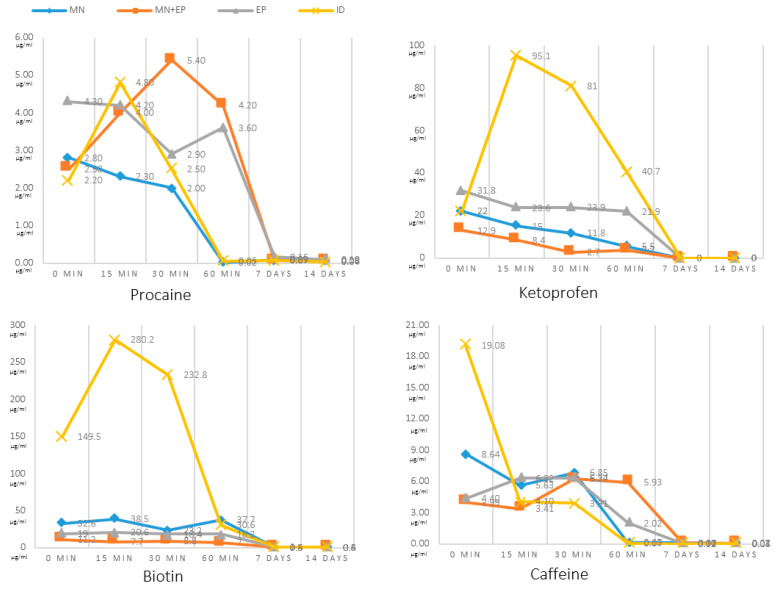
Values of measured cutaneous concentrations (vertical axis), achieved with the HPLC technique for procaine (*n* = 1), ketoprofen (*n* = 1), biotin (*n* = 1), and caffeine (*n* = 1). Concentrations were measured immediately after administration (T0), after 15 min (T15), after 30 min (T30), after 60 min (T60), after one week (D7), and after two weeks (D14). Values are expressed as µg/mL.

**Figure 7 pharmaceutics-13-02016-f007:**
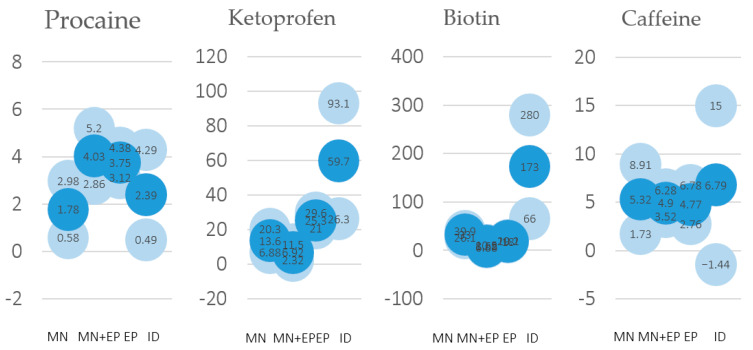
Average values of the HPLC measurements (*n* = 4 for each molecule and disruption technique) during the first hour after drug application (blue bubbles), including standard deviation values (light blue bubbles), for procaine, ketoprofen, biotin, and caffeine.

**Figure 8 pharmaceutics-13-02016-f008:**
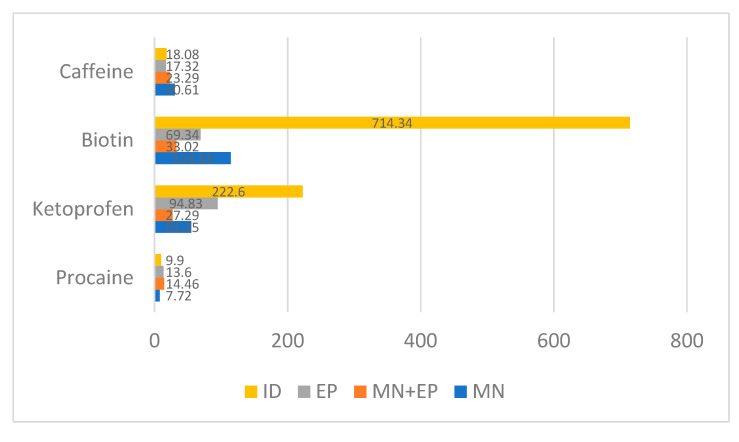
Area under the curve comparing the different molecules and procedures employed. Values are expressed as µg·min/mL.

**Table 1 pharmaceutics-13-02016-t001:** Composition of the different gelified formulations of the study.

Active Ingredient	Methocel E4M	Kathon CG	Water P.CSP 100	Sodium Benzoate	pH
Clorhydrate procaine 2%	1.60%	0.10%	96.30%		5.4 (5.1–5.8)
Ketoprofen 5%	1.60%	0.10%	93.30%		6.6 (6.2–7.0)
Biotin 0.5%	1.60%	0.10%	97.80%		6.7 (6.3–7.1)
Anhydrous caffeine 3%	1.60%	0.10%	92.30%	3%	7.4 (7.1–7.7)

**Table 2 pharmaceutics-13-02016-t002:** Chromatography parameters of the different substances.

	Procaine	Ketoprofen	Biotin	Caffeine
Active ingredient	50 µL	50 µL	50 µL	50 µL
Water		950 µL	950 µL	425 µL
2.5 pH CH_2_O_2_ water	950 µL			
33% Ammonia		20 µL	20 µL	
Methanol				425 µL
Dichloromethane	300 µL	300 µL	300 µL	
Posterior dilution	1:1	1:10	1:10	1:20
Calibration parameters	5–1000 ng/mL	5–5000 ng/mL	5–2000 ng/mL	1–1000 ng/mL

**Table 3 pharmaceutics-13-02016-t003:** Values of epidermal melanocytes and Langerhans cells in the different samples, expressed as the mean number per mm. Standard deviation (σ) is expressed after the mean number in parentheses.

Melanocyte Density				Langerhans Cell Density				
**MN**	**MN + EP**	**EP**	**ID**		**MN**	**MN + EP**	**EP**	**ID**
**3.3** (2.52–4.08)	**5.1** (4.56–5.64)	**4.2** (3.33–5.07)	**4** (3.23–4.77)	T60	**1.2** (0.33–2.07)	**2.1** (1.16–0.94)	**1.3** (0.66–1.94)	**1.1** (0.4–1.8)
**4.3** (3.84–4.76)	**8.4** (7.48–9.32)	**11.3** (9.88–12.72)	**3.1** (2.4–3.8)	D7	**2.2** (1.8–2.6)	**5.3** (4.11–6.49)	**6.8** (5.93–7.67)	**1.7** (0.92–2.48)
**4.6** (4.11–5.09)	**8.1** (6.48–9.72)	**9.4** (8.38–10.42)	**5.2** (4.6–5.8)	D14	**1.4** (0.74–2.06)	**6.4** (5.29–7.51)	**4.1** (3.27–4.93)	**2.2** (1.45–2.95)

**Table 4 pharmaceutics-13-02016-t004:** Variation in the values of melanocyte and Langerhans cell density.

Variation in Melanocyte Density (T0–D7)				Variation in Langerhans Cell Density (T0–D7)			
**MN**	**MN + EP**	**EP**	**ID**	**MN**	**MN + EP**	**EP**	**ID**
+30.30%	+64.71%	+169.05%	−22.5%	+83.33%	+152.38%	+423.08%	+54.54%%

**Table 5 pharmaceutics-13-02016-t005:** Values of dermal and epidermal thickness, expressed as variation over the modal value.

Variation in Epidermal Thickness (T0–D14)			Variation in Dermal Thickness (T0–D14)		
Substance	MN	MN + EP	Substance	MN	MN + EP
Procaine	−14.72%	−5.96%	Procaine	−7.33%	−5.96%
Biotin	−20.87%	+18.77%	Biotin	+31.93%	+0.92%
Caffeine	−30.24%	+39.09%	Caffeine	+12.58%	+4.27%

**Table 6 pharmaceutics-13-02016-t006:** Values of superficial vascular plexus and fibroblast densities. The values are expressed as percentage from the selected area of tissue. Standard deviation values are expressed below the mean number in parentheses.

Superficial Blood Vessel Density					Fibroblast Density				
**MN**	**MN + EP**	**EP**	**ID**			**MN**	**MN + EP**	**EP**	**ID**
**22.2** (1.42)	**25.3** (3.07)	**21.0** (1.48)	**21.6** (2.8)	Procaine	T60	**17.3** (1.42)	**14.7** (2)	**17.1** (2.02)	**17.2** (2.18)
**21.1** (2.34)	**41.2** (4.69)	**22.4** (2.5)	**24.0** (1.95)	Procaine	D7	**16.6** (1.43)	**16.1** (1.42)	**14.3** (1.9)	**16.2** (2.23)
**18.9** (2.21)	**24.3** (2.87)	**24.1** (1.45)	**23.2** (1.4)	Procaine	D14	**17.3** (1.85)	**18.0** (1.9)	**14.9** (1.81)	**15.3** (1.55)
**22.1** (20.53–23.57)	**26.2** (23.9–28.5)	**19.3** (17.45–21.15)	**21.1** (19.52–22.68)	Ketoprofen	T60	**15.5** (14.0–17.0)	**17.4** (15.6–19.2)	**13.7** (12.02–15.38)	**17.2** (14.38–20.02)
**19.6** (18.1–21.1)	**36.3** (32.81–39.79)	**18.3** (17.19–19.41)	**20.6** (19.98–22.22)	Ketoprofen	D7	**18.4** (17.29–19.51)	**14.9** (13.2–16.6)	**16.8** (15.33–18.27)	**20.1** (17.63–22.57)
**18.4** (16.66–20.14)	**25.2** (23.16–27.24)	**4.1** (-3.61–11.81)	**2.2** (-1.48–5.88)	Ketoprofen	D14	**14.6** (12.75–16.45)	**17.3** (15.06–19.54)	**18.2** (17.03–19.37)	**14.7** (12.6–16.8)
**20.3** (18.51–22.09)	**22.7** (20.91–24.49)	**22.8** (20.57–25.03)	**18.2** (16.95–19.45)	Biotin	T60	**18.6** (17.04–20.16)	**16.5** (15.14–17.86)	**14.8** (13.2–16.4)	**16.2** (14.48–17.92)
**24.5** (23.14–25.86)	**41.2** (37.44–44.96)	**17.3** (15.87–18.73)	**22.9** (21.53–24.27)	Biotin	D7	**14.3** (12.62–15.98)	**16.2** (14.2–18.2)	**18.1** (16.8–19.4)	**16.6** (14.91–18.29)
**18.4** (16.95–19.85)	**26.4** (23.98–28.82)	**24.1** (22.29–25.91)	**23.2** (21.42–24.98)	Biotin	D14	**16.1** (14.31–17.89)	**16.2** (14.3–18.1)	**14.9** (13.45–16.35)	**15.2** (13.48–16.92)
**21.7** (3.32)	**24.6** (3.35)	**20.9** (1.81)	**17.9** (1.58)	Caffeine	T60	**17.7** (1.95)	**20.1** (2.62)	**16.3** (1.94)	**15.8** (1.66)
**23.2** (2.3)	**39.8** (3.19)	**24.5** (3.04)	**23.2** (1.6)	Caffeine	D7	**17.2** (1.66)	**15.4** (1.62)	**16.2** (1.47)	**15.9** (1.81)
**21.5** (3.23)	**26.1** (2.88)	**26.3** (1.6)	**27.1** (2.66)	Caffeine	D14	**16.1** (1.73)	**18.3** (1.49)	**15.8** (1.14)	**18.4** (1.28)

**Table 7 pharmaceutics-13-02016-t007:** Variation in the values of superficial blood vessel and fibroblast density.

Variation in Superficial Blood Vessel Density (T0–D7)					Variation in Fibroblast Density (T0–D7)			
Substance	MN	MN + EP	EP	ID	MN	MN + EP	EP	ID
Procaine	−4.95%	+62.85%	+6.67%	+11.11%	−4.46%	+9.52%	−16.37%	−5.81%
Ketoprofen	−11.31	+38.55%	−5.18%	−2.24%	+18.71%	−14.37%	+22.63%	+16.86%
Biotin	+20.69%	+81.50%	−22.30%	+25.82%	−23.18%	−1.81%	+22.23%	+2.47%
Caffeine	+6.91%	+61.79%	+17.22%	+29.61%	−2.82%	−23.38%	−0.61%	+0.63%

## Data Availability

The data that support the findings of this study are available from the corresponding author upon reasonable request.

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
