# Peer review of "Transdermal Drug Delivery in the Pig Skin"

_pharmaceutics, 2021, doi:10.3390/pharmaceutics13122016_

Round 1

Reviewer 1 Report

COMMENTS TO THE AUTHORS

In this manuscript titles “”, the authors aimed to compare different techniques for drug/substances’ transdermal delivery. They used microneedling (MN), electroporation (EP), a combination of both (MN+EP) and intradermal injection (ID) to deliver four difference substances such as ketoprofen, biotin, caffein and procaine to Swine pigs, who were biopsied at different time points. Then, the authors performed a list of assays to deliberate which technique is more efficient for transdermal delivery as well as to evaluate the invasiveness in the skin. This experimental manuscript has been well design, and present potential significance in the drug delivery field as well as medical and pharmacological industry. However, the authors should deeply work on improving the manuscript. The introduction is good, the aims need to be clearer and matching with the conclusions, the material and method section would need to be further disclosed (I would suggest adding a graphical abstract for each technique employed so is better understood where the substance is delivered across the skin). Results are poorly presented, with no statistical analysis, unmatched graph styles, no scale bars, confusing images, and additional data non-relevant to the main content (calibration curves). All this difficult the results interpretation and discussion.

MINOR COMMENTS

L38. Defense OR hydration

L94. THROUGH the pores

L102. Delete a WITH

L105. Molecules THROUGH

L110. Penetrates THROUGH

L114-115. THE more diameter …, THE higher

L115. Delete RESPECTIVELY, because is nor respectively if the abrasion goes before the injection, right?

L138. THE active ingredient… at a concentration of 50 mg/ml

L141. The concentration of sodium biotin was 5 mg/ml …

L145. AT a concentration of 10 mg/Ml.

L151-152. FEMALES and MALES

L151. Add the age of the pigs.

L153. The DESCRIBED experimental procedures, …THE PIGS WERE…

L154, 155 and 156. LATER, … THEN, …. FINALLY, (Review all the initial connectors of the manuscript).

L150. Edit the 2.2 section as: 2.2. Animal model and experimental procedure

L223. Reformulate the sentence as … The times T60, D7 and E14 WERE CHOSEN AS …., SINCE the histological events NEEDED …

L243. Review the superscript of mm2

After the results section, I have stopped the grammar and typo revisions. A deep English grammar revision must be performed for this manuscript.

MAJOR COMMENTS.

Figure 1. Add a letter to each diagram and/or image and an explanation of each on of them in the figure caption. Make sure to add the full name of each procedure and substance applied that match with the acronyms in the picture.

L169. Specify the content of the biopsies (e.g. which skin layers were taken).

L176. If animals were returned to Oviedo University after the experimental procedure, specify (at some point) where the procedures were done.

L184. The speed of what? … of the Injection? … of a shaker?

L190. What are the exact units/settings of medium frequency? What was the time interval in the alternate current?

L197. Table 1 would need to be restructured with 4 rows corresponding the four substances and its %, and 5/6 columns corresponding the gelified components so in each box only appears the % value and the names of Methocel, Kathon, water and pH are not repeated 4 times.

L210. Which is the concentration or % of the saline solution?

L220. Adapt table 2 as recommended for table 1.

L 230. Reformulate the sentence and explain why the EnViSION method was used for.

L235. Clarify the objective and magnification used for revelation.

L246-247. Where the samples blinded as well?

L250. Was all this quantification, in situ or from images, if so how many images per sample were taken and how many cells were counted in total for each stain.

L253. Results are medium or mean?

L256. Figure 2. Again, add letters to each image and indicate/describe in the captions what is depicted.

L262. There was any Statistical analysis performed along the manuscript?

Figure 3. Change color of dashed circles and letters inside since cannot be well distinguished.

L280. What is the color of the vimentin? Dark or clear brown, or beige…

L285. How do you localize/observe inflammatory changes in the sample sections?

Figure 4. I would HIGHLY recommend adding scale bars to all the pictures. Indicate in the figure captions what are pointing the black arrows at, and what is image E.

L296. Add the letter after melanocytes and Langerhans cells that correspond to the image in figure 5 to facilitate the connection.

L299. Briefly explain what the literature [65] found.

Figure 5. Honestly, is difficult to associate the description made in the figure’s caption with what is shown in the image, mainly because of the complexity of the skin structure with many layers, different colors intensity and high cell density. When talking of epidermal basal layers (for example), mark them with a colored dashed line in the picture and indicate it in the caption. Same when talking about all the other structures and cells localizations.

L308. Is there any statistic performed to study differences between treatments/ experimental procedures? Could be this data be plotted in a graph for better representation?

L312. Why the authors did not quantify the substances after the following hours (e.g. 90, 120, 240 min..) or days (1, 2, 3 after procedure)?

FIGURE 6. Indicate the axis of graph A. Are there any error bars? How is the data represented as mean±SD or SEM?

L313. On which do the authors rely to affirm/suggest that the best technique is MN+EP? Compared to what? And what would be the basis to affirm so… to inject as much substance as possible, to last longer in the site, to have an even spread…

L321. All the calibration curves are secondary information and could be either not added in the manuscript or used apart in the supplemental information.

I strongly suggest harmonizing all the graphs “A” from figures 6, 7, 8 and 9 in ONE figure, as well as the format.

Statistical analysis must be performed for HPLC data and superficial vascular plexus and fibroblast density values.

Divide the results in sub-section 3.1, 3.2, etc, according to the assay or technique performed.

Author Response

Dear reviewer 1,

We are truly grateful for your exhaustive revision. All the comments were properly addressed.

The manuscript was corrected following all the minor comments, mostly regarding grammar issues. Thank you very much for your time, a grammar revision was performed, identifying several new errors. Age of the pigs is now included.

Major comments regarding were also addresed and are highly appreciated. I would like to make individual responses for them:

Figure 1. The required letter was added, and both figure legend and main text were improved.

L169. We added the content of the biopsies, but we felt it was more appropiate to include it in the previous sentence describing the 8 mm punch (which is a really wide biopsy, as long as human cutaneous biopsies are usually performed with 3 or 4 mm diameter ones).

L176. Probably the manuscript did not reflect the actual course of the pigs, we modified the text to emphasize that the pigs lived in the farm, going to Oviedo University only to perform the different experimentation procedures described.

L184. The referred speed is about a shaker, we included this point in the text.

L190. We were referring to the general electrical supply in Spain; we clarified this point as it was certainly expressed in a snobby way.

L197. Table 1 was restructured according to your indications, it is much clearer now.

L210. Concentration is the usual 0,9%.

L220. We made the required modifications in the table 2 too.

L 230. The sentence about EnVision method was moved and reformulated, we also modified the subtitle.

L235. We made the required precision about magnification.

L246-247. Both the samples and the observers were blinded.

L250. Quantification was performed through images, it is now described in the text.

L253. Results are referred as mean values, this is a thypo, please excuse the fault.

L256. Figure 2 was clarified, improving the resolution of the text in the upper bars, including a scale bar and explaining the figure both in the legend and also in the text.

L262. Due to the nature of the animals of experimentation, which involved the need of permissions directly from the regional Agriculture and Livestock Ministry, the number of subjects was reduced, limiting any statistical analysis. We conducted several measurements, and in all the performed tables and comparisons the numbers are presented as mean values, we included the standard deviation when several measures were performed and compared areas under the curves.

Figure 3. Colors in Figure 3 were changed to improve legibility, we also included scale bars.

L280. We clarified about vimentin color in Figure 3 legend.

L285. The main inflammatory elements are lymphocytes and granulocytes; both of them have clear morphological features that easily distinguish them (lymphocytes are small roud cells with very scant cytoplasm and granulocytes have a small size with a characteristic polymorphonuclear appearance).

Figure 4. We added scale bars in the figures, and make some additional explanation in the figure.

L296. We added the letter.

L299. The whole paragraph have been explained better, with a more specific mention to the citation.

Figure 5. It is true that the haematoxylin counterstain is weak, and for this reason it is certainly difficult to understand the image. For this reason we added a line delineating the dermoepidermal junction and also included two bars to help identify epidermis and the basal cell layer.

L308. We included a graph comparing the different areas under the curves. The manuscript is much clearer now.

L312. I´m afraid that the institutional permission involved several restrictions concerning the animal welfare. This is the reason for the great time lapses between biopsies, as long as we would also liked to perform more biopsies.

FIGURE 6. We precised in the legend what is the vertical axis, as long as the horizontal axis and the employed units for the measurements are already explained. The data refer to measurements performed in the biopsies taken. Each point in the table corresponds to a single measurement. This is the reason to include the bubble graph at the end of this subsection (Figure 7), where SD is illustrated.

L313. We added some text to explain both HPLC figures.

L321. Calibration curves will be included as supplemental information.

The format has been modified, it´s better displayed this way.

Regarding HPLC data, the presented values correspond to each of the measurements; for this reason any statistical analysis is limited by the reduced sample. However, we joined together the measures from the first hour to offer some statistical data including SD. We also calculated areas under the curve, including a new graph. Regarding melanocyte, Langerhans cell, fibroblast and blood vessel densities, we included SD values in the table.

We are not sure about this last commentary; we included the dendritic cell study in a separate sub-section and moved the morphometrical analysis before the HPLC.

Reviewer 2 Report

This work by Ordiz et al study the in vivo transdermal delivery using commercially available techniques including microneedles, electroporation and intradermal injection. In general, the results obtained here are expected, as many papers have shown the kinetics of the different delivery platforms. Thus the authors don’t make a clear point of innovation or novelty for their work. The main limitation of this work is its poor scholarly presentation and figure description. For instance, figure 2 includes multiple images and analysis that is not described in the caption, the scales are not readable and have no units and for this case, figure 2 is not described in the main text. Moreover, most graphs don’t include error bars and images don’t include scale bars. The authors likely master state of the art, but based on the lack of novelty and poor scholarly presentation, I don’t support publication of this work at the current stage,

Author Response

Dear reviewer 2,

Thank you very much for your critical review of the manuscript. It is true that the pharmacological results are not surprising, but our study is not limited to this aspect. We think the value of the presented work in the literature is about the morphological changes in the skin after performing epidermal disruption procedures. In this regard, we present and try to make an explanation about various morphological features that were not properly addressed before.

I think we have improved the scholarly presentation thanks to reviewers´appreciations. This way several points were issued, particularly including figures and graphs, but also the main text. Figure 2 was poorly presented, but we find it well explained now. Regarding graphs and tables, we included the measurements regarding each technique and active principle. When possible we also included the standard deviation when applicable. Moreover, a table with areas under the curve is now included. A deeper statistical anaylsis is limited by the small sample.

Round 2

Reviewer 1 Report

  1. In tables 3 and 5, express the values as mean±SD instead of putting the SD between parenthesis.
  2. The Y and X-axis in figure 6 must be added in the graphs with their respective units in parentheses even if they are cited in the figure's caption. Two graphs have symbols at every data point (a and d) while the two others don't (b and c). Try to make all four graphs look the same. 
  3. Noticing that not all the figures have enough replicates for statistical analysis, it should be convenient to indicate the "n" of each experiment whenever is possible in each graph's caption as (n=1 or n=2).
  4. In figure 6, all the graphs present different Y-axis units, from 6 to 100 or even 300. (1) I would suggest if the authors can harmonize all the graphs to the same unit ...300. To see differences in those with low substance detected such as in graph 6.a, authors may want to divide the Y-axis into two sections (the bottom from 0 to 8 and the upper from 100 to 300?) (See attached image); (2) authors should also discuss the big differences on concentrations found in each treatment along the time. Why for procaine the maximum is around 6 and for biotin is approx 280?; and (3) You may want to add the title on each graph to easily recognize which treatment correspond.
  5. For future research articles, I suggest to the authors use GraphPad Prism 8 as an easy and straightforward software for simple and complex statistical analysis as well as graphs and figures editor.
  6. L368. The authors state that no significant differences were found regarding fibroblast density. Did the authors finally perform the statistical analysis? If so, please indicate it in the Material and methods section. Same in the conclusions (L 511). If this conclusion is not stated by any statistical result, the authors should avoid using "significant" in the sentence.

Author Response

Dear reviewer 1,

Thank you very much for your revision. Your comments helped decisively to improve the text.

We addressed the 6 points mentioned:

  1. We modified the tables according to your indications.
  2. Concentration units were added to the vertical axis in the table, and time references were included in days and minutes instead of the previous names in the horizontal axis. We changed colours to match figure 6, figure 8 and various tables. We also homogenized the symbols.
  3. We included the n number in figures 6 and 7. We think it is not applicable in figure 8.
  4. We disagree with the first suggestion, as long as harmonizing reference values in the Y axis would induce a bias in the graphs, flattening and heightening the curves artificially. Regarding the second point, the applied molecules have different concentrations, pharmacokinetics and chemical structure and this was a major reason to choose them; this point was mentioned in materials and methods, and we now mention it in the discussion. We applied the third suggestion about the titles in each graph, and we find it helps to understand the figure.
  5. We didn´t know about this software. It seems much better than our PowerPoint figures. We are truly grateful for your comprehension and suggestions.
  6. We modified both points, and also included a pair of new tables to better illustrate the morphological differences found.

Best regards

Reviewer 2 Report

The authors improved the figure description which makes is easier to evaluate the work done. I think now the presentation is clearer the work could find a audience in Pharmaceutics thus I support publication of this work

Author Response

Dear reviewer 2,

Thank you very much for your performing the review of the manuscript and for your support after the revision. Your comments helped decisively to improve the text.

Best regards